# A Diagnostic Accuracy Study of Targeted and Systematic Biopsies to Detect Clinically Significant Prostate Cancer, including a Model for the Partial Omission of Systematic Biopsies

**DOI:** 10.3390/cancers15184543

**Published:** 2023-09-13

**Authors:** Juan Morote, Natàlia Picola, Jesús Muñoz-Rodriguez, Nahuel Paesano, Xavier Ruiz-Plazas, Marta V. Muñoz-Rivero, Anna Celma, Gemma García-de Manuel, Ignacio Aisian, Pol Servian, José M. Abascal

**Affiliations:** 1Department of Urology, Vall d’Hebron Hospital, 08035 Barcelona, Spain; ana.celma@vallhebron.cat; 2Department of Surgery, Universitat Autònoma de Barcelona, 08193 Bellaterra, Spain; 3Department of Urology, Hospital Universitari de Bellvitge, 08907 Hospitalet de Llobregat, Spain; npicola@bellvitgehospital.cat; 4Department of Urology, Hospital Universitari Parc Taulí, 08208 Sabadell, Spain; jmunoz@tauli.cat; 5Clínica Creu Blanca, 08034 Barcelona, Spain; nahuel.paesano@creublanca.es; 6Department of Urology, Hospital Universitari Joan XXIII, 43005 Tarragona, Spain; xruiz.tarragona.ics@gencat.cat; 7Department of Urology, Hospital Arnau de Vilanova, 25198 Lleida, Spain; mvmunoz.lleida.ics@gencat.cat; 8Department of Urology, Hospital Universitari Josep Trueta, 17007 Girona, Spain; gemmagarcia.girona.ics@gencat.cat; 9Department of Urology, Hospital Clinic, 08036 Barcelona, Spain; iasiain@clinic.cat; 10Department of Urology, Hospital Germans Trias i Pujol, 08916 Badalona, Spain; pservian.germanstrias@gencat.cat; 11Department of Urology, Parc de Salut Mar, 08003 Barcelona, Spain; jabascal@psmar.cat; 12Department of Medicine and Health Science, Universitat Pompeu Fabra, 08003 Barcelona, Spain

**Keywords:** clinically significant prostate cancer detection, systematic biopsy, targeted biopsy, prostate biopsy concordance

## Abstract

**Simple Summary:**

Targeted biopsies of suspicious lesions on multiparametric magnetic resonance imaging (mpMRI) improve the sensitivity of systematic biopsies for clinically significant prostate cancer (csPCa) detection. However, systematic biopsies are still recommended, since a small, albeit significant, percentage of csPCa is only detected in them. In a contemporary series, 13.9% of csPCa cases were only detected in systematic biopsies, which represented 6.7% of its overall detection rate. The finding of independent predictive factors for csPCa detection in targeted biopsies allowed the possibility to develop a predictive model for avoiding systematic biopsies with a low risk of missing csPCa.

**Abstract:**

The primary objective of this study was to analyse the current accuracy of targeted and systematic prostate biopsies in detecting csPCa. A secondary objective was to determine whether there are factors predicting the finding of csPCa in targeted biopsies and, if so, to explore the utility of a predictive model for csPCa detection only in targeted biopsies. We analysed 2122 men with suspected PCa, serum PSA > 3 ng/mL, and/or a suspicious digital rectal examination (DRE), who underwent targeted and systematic biopsies between 2021 and 2022. CsPCa (grade group 2 or higher) was detected in 1026 men (48.4%). Discrepancies in csPCa detection in targeted and systematic biopsies were observed in 49.6%, with 13.9% of csPCa cases being detected only in systematic biopsies and 35.7% only in targeted biopsies. A predictive model for csPCa detection only in targeted biopsies was developed from the independent predictors age (years), prostate volume (mL), PI-RADS score (3 to 5), mpMRI Tesla (1.5 vs. 3.0), TRUS-MRI fusion image technique (cognitive vs. software), and prostate biopsy route (transrectal vs. transperineal). The csPCa discrimination ability of targeted biopsies showed an AUC of 0.741 (95% CI 0.721–0.762). The avoidance rate of systematic prostate biopsies went from 0.5% without missing csPCa to 18.3% missing 4.6% of csPCa cases. We conclude that the csPCa diagnostic accuracy of targeted biopsies is higher than that of systematic biopsies. However, a significant rate of csPCa remains detected only in systematic biopsies. A predictive model for the partial omission of systematic biopsies was developed.

## 1. Introduction

Prostate cancer (PCa) is the leading malignant tumour in men from industrialised countries [1]. The European Randomised Screening for Prostate Cancer (ERSPC) trial has shown a persistent decrease in PCa-specific mortality over more than 20 years of follow-up due to the early detection and treatment of clinically significant PCa (csPCa) in the Goteborg section of ERSPC [2]. The early detection of PCa has evolved towards csPCa due to the spread of multiparametric magnetic resonance imaging (mpMRI) and targeted biopsies of suspicious lesions [3]. The high negative predictive value of mpMRI has allowed for a reduction in the number of unnecessary biopsies [4], and targeted biopsies have increased the sensitivity of systematic biopsies for csPCa [5]. However, systematic biopsies remain the only way to biopsy men with high risks of csPCa and negative mpMRI, with some csPCa cases only detected in systematic biopsies still being mpMRI-positive [6,7]. A recent study based on the Goteborg prostate cancer screening trial suggested that targeted biopsies are sufficient in men with a prostate imaging report and data system (PI-RADS) score of 3 or higher [8], while 20% of csPCa cases are missed [9]. In contrast, other studies have suggested that targeted biopsies could be avoided in men with serum prostate-specific antigen (PSA) levels over 10 ng/mL and an abnormal digital rectal examination (DRE) result in order to decrease the demand for mpMRI, given that almost all csPCa cases are detected in systematic biopsies carried out in these men at high risk of csPCa [10,11]. The transperineal route has recently been recommended for reducing the infectious complications of the transrectal route [12,13,14,15]. Systematic biopsy currently includes twelve needle sticks, often detecting insignificant cancer lesions, causing pain and other adverse effects. Targeted biopsies include few needle sticks, and they detect more significant cancer lesions and fewer insignificant cancer lesions than systematic biopsies [6,7,16].

The Barcelona predictive models and risk calculators were developed to predict the risk of csPCa detection just after PCa suspicion to avoid unnecessary mpMRI exams and before mpMRI to improve candidate selection for prostate biopsy. Both predictive models were developed at our centre and externally validated in the metropolitan area of Barcelona [17,18]. Currently, both models are being validated in Catalonia, a country with 7,900,000 inhabitants, in a prospective trial. The main objective of this study is to analyse the accuracy of targeted and systematic biopsies for the detection csPCa in a current population of suspected PCa. We also aim to identify independent predictive factors for the detection of csPCa in targeted biopsies and to develop a predictive model for the partial omission of systematic biopsies.

## 2. Materials and Methods

### 2.1. Design, Setting, and Participants

Among 3.557 men with suspected PCa, due to serum PSA > 3 ng/mL and/or a suspicious DRE result, enrolled in a prospective trial for the validation of the Barcelona risk calculators [17,18] at 10 centres in Catalonia (Spain), 2122 men underwent TRUS-MRI fusion targeted biopsy and systematic prostate biopsy consecutively between 1 January 2021 and 31 December 2022.

### 2.2. MRI and Prostate Biopsy

MpMRI exams were always performed at each participant centre and reported with PI-RADS v.2.0 by an experienced radiologist. A pelvic phased-array surface coil was always used as a 1.5 or 3.0 Tesla MRI magnetic field strength.

MRI-TRUS fusion images were obtained using the cognitive or software technique. TRUS-assisted prostate biopsy was performed via the transrectal or transperineal route. Targeted biopsies of suspected lesions (PI-RADS ≥ 3) were performed, obtaining between 2 and 6 cores, and 12-core TRUS systematic prostate biopsy was also conducted. Prostate biopsies were performed by an experienced urologist at each centre.

Biopsy material was analysed in the pathology department of each participant centre by an experienced pathologist using the International Society of Urologic Pathology grade group (GG) to classify PCa. CsPCa was reported when GG was 2 or higher [19].

### 2.3. Outcome Variable

CsPCa detection in systematic and targeted biopsies was the outcome variable for this study.

### 2.4. Candidate Predictive Variables for the Concordance Degree of csPCa Detection between Systematic and Targeted Biopsies

The candidate predictive variables for the concordance between the systematic and targeted biopsies were age (years), serum PSA level (ng/mL), the type of biopsy (initial vs. repeat), PCa family history (no vs. yes), DRE (normal vs. abnormal), prostate volume (mL), PI-RADS v.2 score (3 vs. 4 vs. 5), the magnetic field strength of mpMRI (1.5 vs. 3 Tesla), TRUS-MRI image fusion technique (cognitive vs. software), and the type of biopsy route (transrectal vs. transperineal). If independent predictive variables were found, the clinical utility of a predictive model developed to predict csPCa in targeted biopsies could be ve-rified.

### 2.5. Statistical Analysis

A statistical analysis was carried out through the harmonisation of the anonymised data sets provided by each participant centre. Quantitative variables are defined as medians and interquartile ranges (IQR: 25–75 percentile). Qualitative variables are defined as percentages. The recodification of quantitative to qualitative variables was performed to estimate the complementarity of the systematic and targeted biopsies. Odds ratios (ORs) and 95% of confidence intervals (CIs) were calculated. The concordance degree of the systematic and targeted biopsies was assessed with the Pearson chi-square test. Independent predictive variables of the concordance between the systematic and targeted biopsies were assessed in a binary logistic regression analysis. A predictive model for the ability to detect csPCa in targeted biopsies was developed from the identified independent predictors of csPCa detection in targeted biopsies. A receiver operating characteristic (ROC) curve was generated, and the area under the curve (AUC) was calculated to assess the discrimination ability of targeted biopsies to detect csPCa. The corresponding specificities to 100, 97.5, and 95% sensitivity threshold likelihoods were assessed, and the rates of avoided unnecessary systematic biopsies for csPCa were undetected. Significant differences were considered when *p* values were lower than 0.05. SPSS v.25 was used to perform this statistical analysis.

## 3. Results

### 3.1. Characteristics of the Study Cohort

The characteristics of the men with suspected PCa selected for this study are presented in Table 1. We found a median age of 68 years and a median serum PSA level of 7.3 ng/mL. The percentage of men with an abnormal DRE was 26.9%, that of repeat prostate biopsies was 33.4%, and that of PCa family history was 7.3%. The median prostate volume was 53 mL. mpMRI was performed with a 3 Tesla scan in 70.5% of cases, whereas for 29.5% of cases, we performed a 1.5 Tesla scan. The percentage of men with a PI-RADS score of 3 was 22.9%, and the percentages of those with PI-RADS scores of 4 and 5 were 54% and 23.1%, respectively. The TRUS-MRI image fusion technique used for targeted biopsies was cognitive in 42.3% of cases, and the software technique was used in 57.7%. The route of prostate biopsy was transperineal in 60.6% of cases and transrectal in 39.4%.

PCa was detected in 1424 men (67.1%), with csPCa in 1026 (48.4%) and insignificant PCa in 398 (18.7%).

### 3.2. Analysis of Concordance between Systematic and Targeted Biopsies according to the Detection of csPCa

The overall concordance–discordance between the systematic and targeted biopsies reporting csPCa is presented in Table 2. Among 2122 suspected PCa cases, the concordance of the systematic and targeted biopsies was observed in 1613 (76%). In 517 cases (24.4%), both biopsies detected csPCa, whereas in 1096 cases (51.6%), csPCa was not found at all, with *p* < 0.001. Discordance between the results reported in the systematic and targeted biopsies was observed in 509 cases (24%). CsPCa was detected only in targeted biopsies in 366 cases (17.3%), whereas it was only detected in systematic biopsies in 143 cases (6.7%), *p* < 0.001.

Considering the 1026 cases of detected csPCa, in 517 cases (50.4%), csPCa was detected in both types of biopsies. In 366 (35.7%) cases, csPCa was detected only in targeted biopsies, whereas in 143 (13.9%) cases, csPCa was only detected in systematic biopsies. The ISUP-GG of csPCa detected only in targeted biopsies corresponded to GG 2 in 233 cases (60.9%), GG 3 in 68 (18.6%), GG 4 in 51 (13.9%), and GG 5 in 24 (6.6%). The ISUP-GG of csPCa detected only in systematic biopsies corresponded to GG 2 in 99 cases (69.2%), GG 3 in 21 (14.7%), GG 4 in 15 (10.5), and GG 5 in 8 (5.6%). After Recoding the ISUP-GG in high-grade csPCa (GG ≥ 3) resulted in 44 cases (30.8%) being detected only in systematic biopsies and 143 cases (39.1%) being detected only in targeted biopsies, *p* = 0.129.

### 3.3. Univariate and Multivariate Analyses of Candidate Predictive Variables for the Concordance Degree of csPCa in Systematic and Targeted Biopsies

The odds ratios and 95% confidence intervals of the candidate predictive variables for the concordance between the systematic and targeted biopsies are presented in Table 3. A univariate analysis showed significant associations between the results of the two types of biopsies and age (≤68 >years), *p* < 0.001; prostate volume (≤53 >mL), *p* < 0.001; the MRI Tesla scanner (1.5 vs. 3.0 Tesla); PI-RADS score (3 vs. 4 vs. 5), *p* < 0.001; the TRUS-MRI image fusion technique (cognitive vs. software), *p* < 0.001; and biopsy route (transrectal vs. transperineal), *p* < 0.001.

The binary logistic regression confirmed that age (ref. ≤68 years) presented a concordance OR of 1.480 (1.182–1.803), *p* < 0.001. Prostate volume (ref. ≤53 mL) presented an OR of 0.582 (0.480–0.720), *p* < 0.001. The MRI Tesla scan (ref. 1.5 Tesla) presented an OR of 1.463 (1.148–1.865), *p* = 0.002. The PI-RADS score (ref. 3) presented an OR of 1.590 (1.352–1.870), *p* < 0.001. The TRUS-MRI image fusion technique (ref. cognitive) presented an OR of 1.333 (1.046–1.701), *p* = 0.020. The biopsy route (ref. transrectal) presented an OR of 1.335 (1.052–1.694), *p* = 0.018. Age and prostate volume were clinical characteristics of suspected PCa in men for predicting concordance between systematic and targeted biopsies. The magnetic field strength of MRI and the PI-RADS score were characteristics of MRI, whereas the TRUS-MRI fusion technique and biopsy route were characteristics associated with prostate biopsy.

The discordance between the systematic and targeted biopsies, according to the findings of the csPCa clinical predictors, is presented in Table 4. We always noted an increasing trend in the rate of csPCa detected in the targeted biopsies but not in the systematic biopsies. However, a significant difference was found in terms of age and prostate volume. The rate of discordance from an age ≤ 68 years to >68 years increased from 20.6 to 27.5%, *p* < 0.001. However, from a prostate volume ≤53 mL to >53 mL, the discordance rate decreased from 29 to 18.9%, *p* < 0.001 (Table 4).

An analysis of the overall discordance between the systematic and targeted biopsies and the detection of csPCa in the systematic or targeted biopsies alone, according to the characteristics of mpMRI and prostate biopsy, is presented in Table 5. The detection of csPCa in the targeted biopsies alone was always significantly higher than that in the systematic biopsies alone. The discordance between the two biopsy types increased from 21 to 25.3% regarding the use of 1.5 vs. 3 Tesla mpMRI, *p* < 0.001; from 11.5 to 31% according to a PI-RADS score of 3 vs. 5, *p* < 0.001; and from 19 to 27.2% according to the transrectal vs. transperineal route and the cognitive vs. software MRI-TRUS image fusion technique, *p* < 0.001.

### 3.4. Development of a Predictive Model for the Discrimination Ability of Targeted Biopsies to Detect csPCa

Due to the findings of predictive variables for the detection of csPCa in targeted biopsies, a predictive model including age, prostate volume, the type of MRI, PI-RADS, the type of TRUS-MRI image fusion technique, and the type of biopsy route was developed, and probabilities of csPCa detection in targeted biopsies were generated. Figure 1 represents the ROC curve of the developed model, showing the discrimination ability of targeted biopsies for csPCa.

The AUC of the ROC curve was 0.741 (95% CI 0.721–0.762). After assessing the thresholds with 100, 97.5, and 95% sensitivity thresholds for csPCa detection, the corresponding specificities were 5.8, 22.5, and 27.4%. Using these parameters, we observed that 11 systematic biopsies were avoided without missing csPCa at 100% sensitivity. In total, 312 systematic biopsies were avoided (14.7%) and 25 csPCa cases were undetected (2.4%) in the targeted biopsies at the 97.5% sensitivity threshold. At 95% sensitivity, 368 systematic biopsies were avoided (18.3%) and 47 csPCa cases (4.6%) were undetected (Table 6).

## 4. Discussion

The present study confirms that targeted biopsies are more sensitive than systematic biopsies for detecting csPCa. However, 6.7% of csPCa cases were detected only in systematic biopsies, whereas 17.2% of csPCa cases were detected only in targeted biopsies. Among 1026 csPCa cases, 13.9% were identified only in systematic biopsies, 35.7% were detected only in targeted biopsies, and 50.4% were identified in both. CsPCa cases detected only in targeted biopsies showed an increasing trend of higher GG, but this was not significant regarding those observed in systematic biopsies. These results are in line with other results previously reported, which supports the need for systematic biopsies [20,21,22,23,24,25,26,27].

The detection of csPCa in systematic biopsies alone is mainly due to non-visible MRI lesions containing this type of tumour [16]. This is the reason for the 9.2% (95% CI 6.9–11.9%) csPCa detection rate in systematic biopsies carried out in men with negative mpMRI (PI-RADS < 3) [4]. Discordance between systematic and targeted biopsies in csPCa detection is defined as the amount of csPCa detected only in one type of biopsy. The reasons for detecting csPCa in only systematic biopsies include the fact that the multifocality of csPCa is not always visible on mpMRI, the fact that there are errors associated with lesion targeting, and the fact that MRI lesions can be missed by radiologists [16].

In 2009, Ahmed et al. suggested that men undergoing systematic biopsy are at risk of the underdiagnosis of csPCa and overdiagnosis of clinically insignificant PCa [28]. In 2017, PROMIS was the trial first providing level-one evidence on the underdiagnosis of csPCa in systematic biopsies. In this trial, a prospective, multicentre, and paired cohort of 576 biopsy-naïve men with suspected PCa was subjected to transperineal 12-core TRUS-guided systematic biopsy; targeted biopsies of the visible lesions on MRI were performed, and prostate mapping biopsy performed every 5 mm served as the reference test. The 48% (95% CI 42–55%) csPCa sensitivity of systematic biopsies increased to 93% (95% CI 88–96%) in targeted biopsies [29]. The PRECISION study was a multicentre and randomised trial, also reported in 2017, including 500 biopsy-naïve men with suspected PCa referred for prostate biopsy. CsPCa was detected in 38% of 245 men who finally underwent MRI and TRUS-guided targeted biopsies, compared to 26% of 235 men who underwent TRUS-guided systematic biopsy. Biopsies were performed evenly via the transrectal or transperineal route. The PRECISION trial generated level-one evidence supporting the notion that targeted biopsies show increased sensitivity compared to systematic biopsies [30]. In 2018, the MRI-First study, a prospective, multicentre, and paired trial, enrolled 251 men with suspected PCa, of whom 53 with PI-RADS < 3 underwent systematic biopsy alone, whereas 198 men with PI-RADS ≥ 3 underwent systematic and targeted biopsies. CsPCa was detected in 29.9% (95% CI 24.3–36%) of systematic biopsies and in 32.3% (95% CI 26.5–38.4%) of targeted biopsies. The percentage of csPCa detected in systematic biopsies alone was 5.2% (95% CI 2.8–8.7%), whereas it was only detected in 7.6% (95% CI 4.6–11.6%) of targeted biopsies, with this difference not being significant. The authors of the MRI-First trial concluded that systematic and targeted biopsies had similar sensitivities for csPCa detection [26]. In 2019, the PAIREDCAP trial included 248 biopsy-naïve men with MRI-visible lesions who underwent systematic biopsy followed by cognitive or software TRUS-MRI fusion targeted biopsy. The csPCa detection rate was 62.1% in systematic biopsies, 46.8% in cognitive TRUS-MRI fusion targeted biopsies, and 60.1% in software TRUS-MRI fusion targeted biopsies, whereas 70% of csPCa cases were detected with all these me-thods. The authors concluded that both systematic and targeted biopsies are required since csPCa was undetected in between 11.5% and 33.3% of cases using any method alone [31].

Brisbane et al. observed that, in 2048 men with suspected PCa and PI-RADS ≥ 3, from two US institutions, 90% (95% CI 89–91%) of the csPCa cases detected were located within a radius of 10 mm from the nearest lesion. In total, 65% (95% CI 63–67%) of csPCa cases were detected within the region of interest identified on mpMRI, whereas 26% (95% CI 24–27%) of csPCa cases were detected in the perilesional area [32]. Recently, Williams et al. retrospectively analysed 41 csPCa cases among 2103 men (1.9%) with suspected PCa, in which csPCa was missed in targeted biopsies and detected in systematic biopsies. The authors reported that 21 csPCa cases (51.2%) were missed in targeted biopsies due to errors in lesion targeting, with 17 (40.5%) being missed due to MRI-invisible lesions and 3 (7%) being overlooked due to MRI lesions missed by the radiologist. They also compared the characteristics of 425 men with suspected PCa in whom csPCa was detected in targeted biopsies and those of 45 men in whom csPCa was missed in targeted biopsies. They reported lower serum PSA levels, higher prostate volumes, lower numbers of core-targeted biopsies, and lower PI-RADS scores in men with undetected csPCa in targeted biopsies, although the multivariate analysis only found an association of lower PI-RADS scores with undetected csPCa in the targeted biopsies [33].

Our study reported predictive variables for the concordance between systematic and targeted biopsies in csPCa detection. We observed that older age and a larger prostate volume increased the discordance between systematic and targeted biopsies, as also observed by Williams et al. [34]. Regarding the characteristics of mpMRI, we reported an increased discordance when the magnetic field strength was 3 Tesla compared with 1.5 Tesla, with this mainly being due to an increased number of csPCa cases detected only in targeted biopsies. Such data have not been reported previously, beyond the finding that csPCa detection is not increased using 3 Tela mpMRI [28]; however, more lesions with csPCa may be detected when a higher magnetic field strength is used. As reported by Williams et al. [34], we also observed how the complementarity between systematic and targeted biopsies increased with the PI-RADS score. We observed that the detection rate of csPCa in targeted biopsies alone increased from 4.9% in men with PI-RADS 3 to 18.8% in those with PI-RADS 4 and to 25.9% in those with PI-RADS 5. However, the rate of csPCa detected in systematic biopsies alone remained stable (5.1–7.5%) from PI-RADS 3 to 5. The reason for this finding may be related to the increased visibility of csPCa detected when the PI-RADS score increases. The PAIREDCAP trial also supports our finding of the higher csPCa sensitivity of software TRUS-MRI image fusion compared to cognitive TRUS-MRI image fusion [24]. Finally, we observed that 10.2% of csPCa cases were only detected in targeted biopsies when the transrectal route was used, while 21.9% were only detected when the transperineal route was used. This percentage of csPCa cases detected only in systematic biopsies decreased from 8.9 to 5.4%. Such data have not been reported previously, beyond the overall increased sensitivity reported for the transperineal route, which is also influenced by the location of the lesions [35].

Because independent predictive factors of the csPCa discrimination ability of targeted biopsies were found (age, prostate volume, the type of MRI, PI-RADS, the type of TRUS-MRI image fusion technique, and the type of biopsy route), a model for the prediction of csPCa in targeted biopsies was developed. The AUC of the model was only 0.741 (95% CI 0.721–0.762); however, the ROC curve presented good performance in the high-sensitivity zone, which allowed for 11 systematic biopsies (0.5%) to be avoided without missing csPCa at 100% sensitivity, whereas 14.7% of systematic biopsies were avoided, missing 2.4% of csPCa at 97.5% sensitivity, and 18.3% systematic biopsies were avoided, missing 4.6% of csPCa at 95% sensitivity. No predictive model for avoiding systematic biopsies in men with suspected PCa and PI-RADS ≥ 3 has been reported previously. We summarise that, currently, targeted biopsies are more sensitive than systematic biopsies for detecting csPCa. However, systematic biopsies are still needed to detect all csPCa cases. The newly developed predictive model could increase the avoidance rate from a small number of systematic biopsies without missing csPCa to up to 18.3% of systematic biopsies missing less than 5% of csPCa.

The main limitation of our study is the definition of csPCa in prostate biopsies, which may not represent the true pathology of the entire prostate gland. The recent recruitment of selected men prevented knowing the evolution of detected tumours. This study is a post hoc analysis of a cohort of men who underwent systematic and targeted biopsies in a prospective trial designed to validate the Barcelona risk calculations in Catalonia. The prediction of the csPCa discrimination ability of targeted biopsies could be improved through other independent predictive variables not included in our study (e.g., the location and size of visible lesions).

Improvements in MRI, based on future increases in magnetic field strength, could enhance the visibility of csPCa. The combination of MRI with Ga^68^PSMA-PET (Ga^68^PSMA-PET-MRI) is promising. The metabolic activity detected in MRI-visible lesions or in areas without visible lesions can improve the effectiveness of targeted biopsies. The specificity of MRI could also be increased by differentiating visible lesions with no or low metabolic activity [35,36]. Additionally, future improvements in TRUS-MRI fusion images and robotic targeted biopsies will also help to avoid systematic biopsies and decrease the current overdetection of insignificant PCa [8,9].

## 5. Conclusions

Targeted biopsies currently remain more sensitive than systematic biopsies for detecting csPCa. However, 13.9% of csPCa cases were only detected in systematic biopsies, which represented 6.7% of the overall csPCa detection rate. The disagreement between the systematic and targeted biopsies for csPCa detection was multifactorial, with age, prostate volume, MRI magnetic field strength, PI-RADS score, TRUS-MRI image fusion technique, and biopsy route being independent predictive variables. The predictive model developed for csPCa detection in targeted biopsies was able to increase the omittance rate of systematic biopsies from 0.5% without missing csPCa to 18.3% missing 4.6% of csPCa cases.

## Figures and Tables

**Figure 1 cancers-15-04543-f001:**
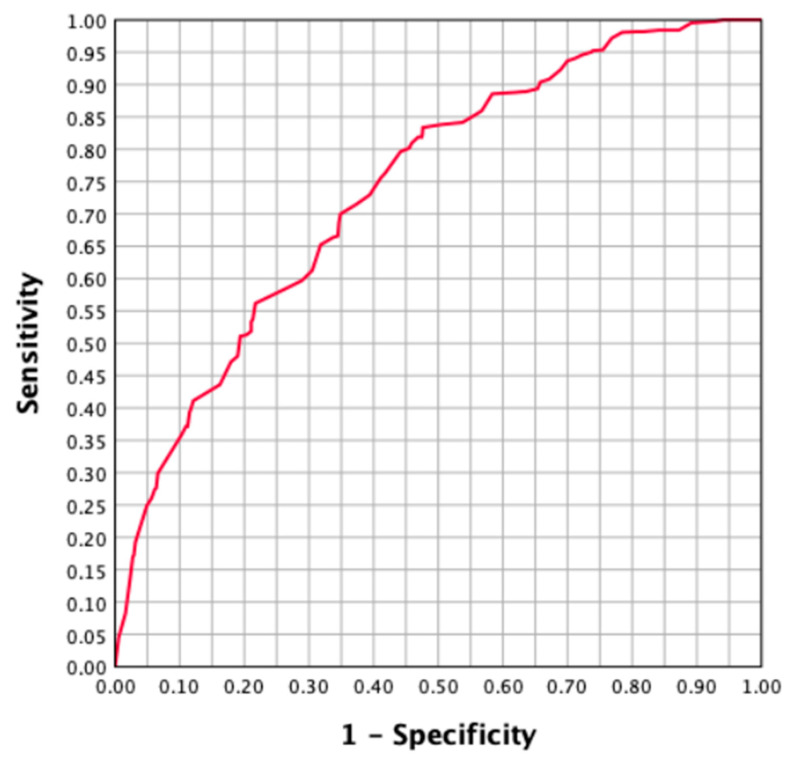
ROC curve showing the csPCa discrimination ability of targeted biopsies.

**Table 1 cancers-15-04543-t001:** Characteristics of men subjected to systematic and targeted biopsies.

Characteristic	Measurement
Number of men, n	2122
Median age, years (IQR)	68 (62–73)
Median serum PSA, ng/mL (IQR)	7.3 (5.2–11.0)
Abnormal DRE, n (%)	570 (26.9)
Repeat prostate biopsy, n (%)	708 (33.4%)
PCa family history, n (%)	154 (7.3)
Median prostate volume, mL (IQR)	53 (39–95)
3 Tesla mpMRI, n (%)	1493 (70.5)
PI-RADS 3, n (%)	485 (22.9)
PI-RADS 4, n (%)	1146 (54.0)
PI-RADS 5, n (%)	491 (23.1)
Software TRUS-MRI image fusion, n (%)	1224 (57.7)
Transperineal route, n (%)	1286 (60.6)
ISUP grade group of detected tumours, n (%)	
1	398 (27.9)
2	522 (36.7)
3	184 (12.9)
4	172 (12.1)
5	148 (10.4)
PCa, n (%)	1424 (67.1)
csPCa, n (%)	1026 (48.4)

PSA = prostate-specific antigen; DRE = digital rectal examination; PCa = prostate cancer; csPCa = clinically significant PCa; IQR = interquartile range; mpMRI = multiparametric magnetic resonance imaging; PI-RADS = prostate imaging report and data system; TRUS-MRI = transrectal ultrasound–magnetic resonance imaging; n = number; ISUP = international society of urologic pathology.

**Table 2 cancers-15-04543-t002:** Analysis of overall concordance–discordance of csPCa detection rate in systematic and targeted biopsies.

Number of Men	Discordance n (%)	CsPCa in Systematic and Targeted Biopsies n (%)			
		− −	− +	+ −	+ +
2122	509 (24.0)	1096 (51.6)	366 (17.3)	143 (6.7)	517 (24.4)
*p* < 0.001					

**Table 3 cancers-15-04543-t003:** Univariate and multivariate analyses of candidate predictive variables for the concordance between systematic and targeted biopsies in detecting csPCa.

Predictive Variable	Univariate Analysis		Multivariate Analysis	
	Odds Ratio (95% CI)	*p* Value	Odds Ratio (95% CI)	*p* Value
Age, ref. ≤68 years	1.464(1.119–1.789)	<0.001	1.480 (1.182–1.803)	<0.001
Serum PSA, ref. ≤7.3 ng/mL	0.972 (0.786–1.186)	0.777	0.893 (0.722–1.105)	0.299
DRE, ref. normal	1.233 (0.989–1.536)	0.066	1.087 (0.856–1.379)	0.493
Type of biopsy, ref. initial	1.112 (0.901–1.371)	0.323	1.176 (0.939–1.473)	0.159
PCa family history, ref. no	1.205 (0.833–1.744)	0.312	1.306 (0.888–1.921)	0.175
Prostate volume, ref. ≤53 mL	0.570 (0.465–0.698)	<0.001	0.582 (0.470–0.720)	<0.001
Type of mpMRI, ref. 1.5 Tesla	1.274 (1.017–1.596)	<0.001	1.463 (1.148–1.865)	0.002
PI-RADS category, ref. 3	1.690 (1.431–1.956)	<0.001	1.590 (1.352–1.870)	<0.001
Type of TRUS-MRI fusion technique, ref. visual	1.483 (1.206–1.823)	<0.001	1.333 (1.046–1.701)	0.020
Prostate biopsy approach, ref. transrectal	1.592 (1.288–1.968)	<0.001	1.335 (1.052–1.694)	0.018

PSA = prostate-specific antigen; DRE = digital rectal examination; PCa = prostate cancer; mpMRI = multiparametric magnetic resonance imaging; PI-RADS = prostate imaging report and data system; CI = confidence interval; TRUS-MRI = transrectal ultrasound–MRI.

**Table 4 cancers-15-04543-t004:** Overall discordance of systematic and targeted biopsies for the detection of csPCa and its detection in systematic or targeted biopsies alone, according to the clinical characteristics of the analysed population.

According to:	Number of Men	Discordance	*p* Value	Only in TB	Only in SB	*p* Value
Age:						
≤68 years, n (%)	1087	224 (20.6)	<0.001	162 (14.9)	62 (5.7)	<0.001
>68 years, n (%)	1035	285 (27.5)		204 (19.7)	81 (7.8)	<0.001
Serum PSA:						
≤7.3 ng/mL, n (%)	1064	258 (24.2)	0.777	191 (18.0)	67 (6.3)	<0.001
>7.3 ng/mL, n (%)	1058	251 (23.7)		175 (16.5)	76 (7.2)	<0.001
DRE:						
Normal, n (%)	1552	356 (22.9)	0.062	268 (17.3)	88 (5.7)	<0.001
Abnormal, n (%)	570	153 (26.8)		98 (17.2)	55 (9.6)	<0.001
Type of biopsy:						
Initial, n (%)	1414	330 (23.3)	0.323	238 (16.8)	92 (6.5)	<0.001
Repeat, n (%)	708	179 (15.6)		128 (18.1)	51 (7.2)	<0.001
PCa family history:						
No	1968	467 (23.7)	0.328	330 (16.8)	127 (7.0)	<0.001
Yes	154	42 (27.3)		36 (23.4)	6 (3.9)	<0.001
Prostate volume:						
≤53 mL, n (%)	1064	309 (29.0)	<0.001	225 (21.1)	84 (7.3)	<0.001
>53 mL, n (%)	1058	200 (18.9)		141 (13.3)	59 (5.6)	<0.001

TB = targeted biopsy; SB = systematic biopsy; PSA = prostate-specific antigen; DRE = digital rectal examination.

**Table 5 cancers-15-04543-t005:** Overall discordance in csPCa detection between systematic and targeted biopsies and its detection in systematic or targeted biopsies alone according to the characteristics of mpMRI and prostate biopsy.

According to:	Number of Men	Discordance	*p* Value	Only in TB	Only in SB	*p* Value
MRI Tesla scan						
1.5 Tesla	625	131 (21.0)	0.035	84 (13.4)	47 (7.5)	<0.001
3.0 Tesla	1497	378 (25.3)		282 (18.8)	96 (6.4)	<0.001
PI-RADS score:						
PI-RADS 3, n (%)	485	56 (11.5)	<0.001	24 (4.9)	32 (6.6)	<0.001
PI-RADS 4, n (%)	1146	301 (26.3)		215 (18.8)	86 (7.5)	<0.001
PI-RADS 5, n (%)	491	152 (31.0)		127 (25.9)	25 (5.1)	<0.001
Biopsy route:						
Transrectal, n (%)	836	159 (19.0)	<0.001	85 (10.2)	74 (8.9)	<0.001
Transperineal, n (%)	1286	350 (27.2)		281 (21.9)	69 (5.4)	<0.001
MRI-TRUS fusion technique:						
Cognitive, n (%)	836	159 (19.0)	<0.001	103 (11.5)	76 (8.5)	<0.001
Software, n (%)	1286	350 (27.0)		263 (21.5)	67 (5.5)	<0.001

TB = targeted biopsy; SB = systematic biopsy; PI-RADS = prostate imaging report and data system.

**Table 6 cancers-15-04543-t006:** Specificities corresponding to the 100, 97.5, and 95% sensitivity thresholds for csPCa detection in targeted biopsies and potential reductions in systematic biopsies and undetected csPCa.

Threshold	Sensitivity (%)	Specificity (%)	Avoided SB (%)	Undetected csPCa (%)
0.038	100	5.8	11 (0.5)	0
0.061	97.5	22.5	312 (14.7)	25 (2.4)
0.071	95.0	27.4	388 (18.3)	47 (4.6)

## Data Availability

The data presented in this study are available on request from the corresponding author.

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
