# Peer review of "A Diagnostic Accuracy Study of Targeted and Systematic Biopsies to Detect Clinically Significant Prostate Cancer, including a Model for the Partial Omission of Systematic Biopsies"

_cancers, 2023, doi:10.3390/cancers15184543_

Round 1
Reviewer 1 Report (New Reviewer)
This study examines the current concordance between systematic and targeted biopsies. For this, they examined 2122 suspected prostate cancer cases with serum PSA>3 ng/ml and a suspicious DRE who underwent systematic and targeted biopsies. They determined the factors predicting the finding of csPCa in targeted biopsies and developed a predictive model for identification of clinically significant prostate cancer in these biopsies. The predictive model was able to avoid systematic biopsies with a very low risk of missing csPCa. The findings have significant implications in clinical management of prostate cancer. The limitations of the study include lack of novelty and defining csPCa in prostate biopsies that may not reflect true biology.
A few corrections are needed to sentences. e.g. line 317
Author Response
Response letter to Reviewer 1
This study examines the current concordance between systematic and targeted biopsies. For this, they examined 2122 suspected prostate cancer cases with serum PSA>3 ng/ml and a suspicious DRE who underwent systematic and targeted biopsies. They determined the factors predicting the finding of csPCa in targeted biopsies and developed a predictive model for identification of clinically significant prostate cancer in these biopsies. The predictive model was able to avoid systematic biopsies with a very low risk of missing csPCa. The findings have significant implications in clinical management of prostate cancer. The limitations of the study include lack of novelty and defining csPCa in prostate biopsies that may not reflect true biology.
Comments on the Quality of English Language. A few corrections are needed to sentences. e.g. line 317.
Response: Thank you for your comment. A new English proofreading has been performed.

Reviewer 2 Report (New Reviewer)
Please see the attachment

no
Author Response
Response letter to Reviewer 2.
We thank the comments and suggestions of the Reviewer, and the opportunity to make modifications in the manuscript according to them. Modifications appears in red color in the reviewed manuscript.
Overall Comment:
The study is a detailed report of a recent protocolized study in Catalonia that includes 2.122 biopsy-naïve patients.
- The study is prospective.
- The study stated eligibility criteria.
- The study gives adequate and OK rationale for using systematic biopsies.
- The study reports well described baseline patient characteristics
- The study gives estimates for the diagnostic performance of the biopsies and the precision (95% CI).
The study made a model to reduce use of combi biopsies (systematic and targeted biopsies)
Conclusion T3 MRI was better than T1.5 MRI. Transperitoneal biopsy was better than transrectal biopsy. Targeted biopsy detected more significant cancers than systemic biopsy. The findings are consistent with many previous studies. The study did not comment on the use of a risk score previously developed in Barcelona by the first author [1].
A revised version of the manuscript should adhere to standards for reporting diagnostic accuracy for MRI guided biopsies [2, 3].
Response: The present study is a retrospective observational analysis of men who underwent targeted and systematic biopsies in the population selected prospectively for the validation of the Barcelona predictive models and risk calculator of csPCa in Catalonia, a Spanish country with 7.900.000 inhabitants. Therefore, the validations of developed model to avoid some systematic biopsies should be done in other population.
Specific Comments:
1.Title: The title is promising. But the title did not identify the study as a diagnostic accuracy study (which is more than the concordance between targeted and systematic biopsies). A descriptive title is “A diagnostic accuracy study of targeted and systematic biopsies to detect significant localized prostate cancer, including a model for a partial omission of systematic biopsies”.
Response: Thank you for this suggestion. The title has been changed according to your suggestion.
- Abstract: should focus on the diagnostic accuracy of the biopsies (and not on concordance between the two types of biopsies). The abstract should conclude whether the study points to the combi or to the targeted biopsy to detect csPCa.
Response: Thank you. We agree. Changes have been made in red color in the abstract (lines 38-54).
- 4. Introduction:
First paragraph should summarize literature. Systematic biopsies include 12+ needle sticks, often detecting insignificant cancer lesions, causing pain and other adverse effects. Literature of patients with negative systematic biopsies showed targeted biopsies can help to detect csPCa. Targeted biopsies include few needle sticks and detect more significant cancer lesions and fewer insignificant cancer lesions than systematic biopsies.
Response: The sentence proposed by the Reviewer has been added in lines 78-81.
Second paragraph should be the submitted Discussion lines 277 to 311.
Response: The second paragraph has been submitted to the current discussion in lines 296-299
Third paragraph should inform of the Barcelona MRI risk calculator (BNC-RC-2)[1]. So the choice is either combi biopsies, or a model of using only targeted biopsies for a group of patients (like the Barcelona risk score). The model implies the authors are willing not to use combi biopsies for all patients. However, that seems not to be the case.
Response: Currently, the Barcelona risk-calculators are in the process of validation in Catalonia. Therefore, a selection of men according to the risk of csPCa seems not appropriate. However, our objective was not to analyse the possible impact of the Barcelona-RC-2 in avoiding systematic biopsies. The Barcelona-RC-2 is able to avoid combi biopsies, but not only systematic biopsies. However, we agree with the Reviewer comment, and we decide to analyse the accuracy of targeted and systematic biopsy for csPCa detected in the subset of men who underwent both biopsies.
Fourth paragraph should give the hypothesis/aims of the study (would the BCN-RC-2 work well for the 10 Catalonian centers or not?).
Response: If the BCN-RC-2 works or not will be assessed when the validation will be concluded, but this was not the objective of this study.
- 19. Patients and Methods:
The study did not include a flow chart of how patients were sent from primary sector for cancer detection (like that used in [4], and how many patients were excluded on the way to the diagnosis of PCa. One excluded group is patients with contradiction against MRI 2 scanning. The study could inform how many patients had only systematic biopsies or targeted biopsies, but not both. The study should report how many patients dropped out. In case, the study should report that a science ethics committee had approved the protocol (as in [4]), and that all patients had accepted to take part in the study after having been informed of the trial. That no patients later withdrew their consent to participate in the study.
Response: The early detection of csPCa in Catalonia is currently an opportunistic program of csPCa early detection, in which PCa suspicion is usually assessed by urologist in the primary sector although some men are referred to urologist due to a serum PSA elevation. Participants in our validation trial of BCN-RCs were men with PCa suspicion in whom pre-biopsy mpMRI was performed, and targeted and/or systematic biopsies. Some men with PI-RADS < 2 , in whom a high risk of csPCa existed and only systematic biopsies were performed. Men selected for present retrospective and observational analysis were those with suspected lesions in whom targeted and systematic biopsies were carried out.
- The study did not report the period for the study.
Response: Yes, the period for the study was from January 1, 2021, to December 31, 2022. This data has been incorporated in current lines 98-99 of the manuscript.
- The study did not tell whether it included only (and all) consecutive patients.
Response: The selected men underwent consecutively targeted and systematic biopsies in the 10 participant centers. This is now specified in current line 98.
10a. The study did not give the number of cores per cancer lesion, whether some MRI detected more than one lesion, length of the cores and percentage with cancer in the cores.
The study did not tell the PiRADS score system used (PIRADS v2 as in a previous study by the first author). The study did not report the experience of the MRI readers (at ten different Catalonian centers). Whether the MRI were read by one or more than one reader, whether the study included a central review of all MRI and all biopsy ISUP.
10b. The study did not use a common protocol for the route for the biopsies, and number of cores per lesion for the systematic and targeted biopsies.
Response: Available data regarding the important aspects suggested by the Reviewer is now incorporated in the point 2.2. of material and method (lines 100-111).
1_2_b_._ _T_h_e_ _s_t_u_d_y_ _d_i_d_ _n_o_t_ _d_e_b_a_t_e_ _t_h_e_ _c_h_o_i_c_e_ _t_o_ _d_e_f_i_n_e_ _c_s_P_C_a_ _(_b_i_o_p_s_y_ _I_S_U_P_ _≥2_)_ _r_e_l_a_t_i_v_e_ _t_o_ _o_t_h_e_r_ _p_u_blished definitions of csPCa. The study did not tell how many patients underwent radical prostatectomy (RP) and had RP specimen ISUP grade. Histologic ISUP grade tends to be higher than the biopsy ISUP grade. How many cores had two different Gleason scores? How did the study report biopsies for patients with biopsies with only one ISUP grade?
Response: Effectively, we do not debate the limitation of using the classification of csPCa as ISUP > 2, and the under-grading observed in radical prostatectomy specimens was not available in our data base. Pathology reported the maximal ISUP in each targeted biopsy and in the systematic biopsy. Limitations of the study are commented in lines 300-396.
13a. 13b. The study did not tell whether clinical data and MRI findings were available for the urologists that made the systematic biopsies. The study did not inform of the clinical course (if any) between systematic and targeted biopsies, or of whether the caretaker for the second type of biopsy knew the findings of first biopsy.
Response: Targeted and systematic biopsies were carried out in the same biopsy session and by the same urologist under general anesthesia. Urologist who performed prostate biopsies in each center had available clinical data in electronic record as well MRI report and images They also performed the segmentation of suspicious lesions when MRI-TRUS fusion targeted biopsies was made.
- The findings on systematic and targeted biopsies were not shown in a Venn diagram.
Response: I´m sorry for the inconvenience, we do not know how to perform the Venn diagram.
- The study did not give the number of undetermined ISUP grade.
Response: Undetermined ISUP grade was not reported by pathologist
- The study did not tell how it reacted to missing data.
Response: Cases with missing data corresponding to the variables required for the Barcelona risk-calculators validation were previously excluded.
- The study did not tell whether the ten centers differed in MRI and biopsies (especially the primary Barcelona center vs the 9 other centers), differed between several readers of MRI, the MRI reviewers and the biopsy performers level of expertise. The study did not tell whether it used central review of MRI and biopsy ISUP scare.
Response: Central review of MRI and pathology was not required. Variations among the 10 participant centers were accepted because we aimed to validate the Barcelona-risk calculator among the variability of the real clinical practice in Catalonia.
- The study did not give the time interval between the two types of biopsies. The study did not tell whether any patient had treatment between first and second biopsy.
Response: All men who had previous negative prostate biopsy did not received any treatment. We only know the type of biopsy (initial vs repeated) but not the time between biopsies.
- Cross tabulation of the biopsies did not report negative findings. Table 5 gave details of 366 patients with only csPCa in targeted biopsy and of 143 patients with only csPCa in systematic biopsy. The table showed PIRADS scores. But the endpoint for the study was biopsy ISUP grade. The table should tell the biopsy ISUP grade 2-5 for the groups of patients with detected csPCa in only one type of biopsy. How many patients had ISUP 5 that was detected only in systematic biopsies?
Response: I believe that the Reviewer is referring to the Table 2. This table does not show the PI-RADS scores. However, I agree with the suggestion of knowing the ISUP 2-5 for the subsets of patients in whom csPCa was detected in only targeted and systematic biopsies. Unfortunately, it is difficult to incorporate this data in the table 2. I incorporated the data in the text in lines 201 to 207. An increased trend of higher ISUP-GG was observed in csPCa only detected in targeted biopsies regarding those only detected in systematic biopsies. However, this increase was not significant.
A comment about this results is now reported in the discussion in lines 342-344.
The protocol for the study may have consent from the patients for registration and monitoring of the post-biopsy phase. If multivariate analyses used overall survival as endpoint instead of csPCa, the p values for the variables will be different, at least later on.
Response: I agree with the Reviewer comment. However, prostate biopsies were made between the years 2021 and 2022, and overall survival cannot was not explored due to the short follow up period of treated patients.
Discussion:
Should start with present draft line 312.
Response: This paragraph has been moved to the current lines 298-305 at the beginning of Discussion.
Discussion should debate the usefulness between combi and targeted biopsies and tell what is new for the debate with regard to two ps papers by the first author [1, 4].
Response: A debate about the usefulness of combi and targeted biopsies has been done in the discussion. This study reports a long series of suspected PCa cases subjected to combi biopsies, analysing the rate csPCa detected only in systematic biopsies in order to know if they are needed in Catalonia. This new information is not not related with our previous papers, except that this observational study was made in the population for the BCN-RCs in Catalonia.
A doctoral thesis by Lars Boesen, Herlev University Hospital, Copenhagen, Denmark, 2021, has a long discussion on targeted vs combined targeted and systematic biopsies.
Response: I love the articles published by Lars Boesen. However, we have not found specific articles from Lars about the utility and need of systematic biopsies. I congratulate Lars Boesen for his wonderful PhD.
- Limitations: The study had hundreds of patients who underwent RP after diagnosis of csPCa. The study did not tell the histologic ISUP grade. The study did not tell management and outcome for the patients after diagnosis of csPCa.
Response: Unfortunately, we have no data about the follow-up of patients with csPCa detected in this validation trial. Our objective in this study was to analyse the need of systematic biopsies in Catalonia.
- Perspective: The study proposed a model but did not give any indication of whether the model is used today in Catalonia or elsewhere. Maybe the clinical outcome will remain unchanged by implementation of the model. Perhaps international cooperation between centers working on the topic will increase the power of future analyses on the topic.
Response: I agree with the Reviewer comments. In the present study we only have developed a model useful for the proof the concept that systematic biopsies could be omitted using a predictive model and the counterpart of missed csPCa. This model needs to be calibrated, adjusted the cut-off value, and validated internally and externally.
I would be pleased to find an international cooperation, specially to validate the Barcelona-risk calculators. We really believe that Barcelona-risk calculator works better than Rotterdam-risk calculator (Morote et al. Comparison of Rotterdam and Barcelona Magnetic Resonance Imaging Risk Calculators for Predicting Clinically Significant Prostate Cancer. Eur Urol Open Sci. 2023 Jul, 53, 46-54.). Therefore, I would be pleased if you want to validate the BCN-RCs in the nice series of Herlev University Hospital. We could collaborate providing the script to calculate the individual probabilities of csPCa based on both Barcelona-risk calculators, doing the calculation in your data base, or any other possibility. If you consider that we could collaborate, please let me know at my e-mail: juan.morote@vallhebron.cat
The study reported many references that are very important for the topic. The study may also discuss other aspects [5-13].
bibliography
[1] Morote J, Borque-Fernando A, Triquell M, Campistol M, Celma A, Regis L, et al. A Clinically Significant Prostate Cancer Predictive Model Using Digital Rectal Examination Prostate Volume Category to Stratify Initial Prostate Cancer Suspicion and Reduce Magnetic Resonance Imaging Demand. Cancers (Basel) 2022;14.
[2] Moore CM, Kasivisvanathan V, Eggener S, Emberton M, Futterer JJ, Gill IS, et al. Standards of reporting for MRI-targeted biopsy studies (START) of the prostate: recommendations from an International Working Group. Eur Urol 2013;64:544-52.
[3] Bossuyt PM, Reitsma JB, Bruns DE, Gatsonis CA, Glasziou PP, Irwig L, et al. STARD 2015: an updated list of essential items for reporting diagnostic accuracy studies. BMJ 2015;351:h5527.
[4] Morote J, Borque-Fernando A, Triquell M, Celma A, Regis L, Escobar M, et al. The Barcelona Predictive Model of Clinically Significant Prostate Cancer. Cancers (Basel) 2022;14.
[5] Petov V, Azilgareeva C, Shpikina A, Morozov A, Krupinov G, Kozlov V, et al. Robot-Assisted Magnetic Resonance Imaging-Targeted versus Systematic Prostate Biopsy; Systematic Review and Meta-Analysis. Cancers (Basel) 2023;15.
[6] Hamdy FC, Donovan JL, Lane JA, Metcalfe C, Davis M, Turner EL, et al. Fifteen-Year Outcomes after Monitoring, Surgery, or Radiotherapy for Prostate Cancer. N Engl J Med 2023;388:1547-58.
[7] Sartor O. Localized Prostate Cancer - Then and Now. N Engl J Med 2023;388:1617-8.
[8] Sountoulides P, Pyrgidis N, Polyzos SA, Mykoniatis I, Asouhidou E, Papatsoris A, et al. Micro-Ultrasound-Guided vs Multiparametric Magnetic Resonance Imaging-Targeted Biopsy in the Detection of Prostate Cancer: A Systematic Review and Meta-Analysis. J Urol 2021;205:1254-62.
[9] Wegelin O, Exterkate L, van der Leest M, Kummer JA, Vreuls W, de Bruin PC, et al. The FUTURE Trial: A Multicenter Randomised Controlled Trial on Target Biopsy Techniques Based on Magnetic Resonance Imaging in the Diagnosis of Prostate Cancer in Patients with Prior Negative Biopsies. Eur Urol 2019;75:582-90.
[10] Kasivisvanathan V, Rannikko AS, Borghi M, Panebianco V, Mynderse LA, Vaarala MH, et al. MRI-Targeted or Standard Biopsy for Prostate-Cancer Diagnosis. N Engl J Med 2018;378:1767-77.
[11] Wegelin O, van Melick HHE, Hooft L, Bosch J, Reitsma HB, Barentsz JO, et al. Comparing Three Different Techniques for Magnetic Resonance Imaging-targeted Prostate Biopsies: A Systematic Review of In-bore versus Magnetic Resonance Imaging-transrectal Ultrasound fusion versus Cognitive Registration. Is There a Preferred Technique? Eur Urol 2017;71:517-31.
[12] Loeb S, Vellekoop A, Ahmed HU, Catto J, Emberton M, Nam R, et al. Systematic review of complications of prostate biopsy. Eur Urol 2013;64:876-92.
[13] Boesen L, Norgaard N, Logager V, Balslev I, Bisbjerg R, Thestrup KC, et al. Assessment of the Diagnostic Accuracy of Biparametric Magnetic Resonance Imaging for Prostate Cancer in Biopsy-Naive Men: The Biparametric MRI for Detection of Prostate Cancer (BIDOC) Study. JAMA Netw Open 2018;1:e180219.
Response: I agree with the Reviewer. There are multiple and important issues to be commented and discussed. However, they must be closely related with the aim of the study.
I thank you very much to the Reviewer for his extensive and deep review carried out. I appreciate this review very much. I was not able or knew how to respond sometimes; however, I believe that his suggestions have improved this manuscript.
Kind regards,
Juan Morote

Reviewer 3 Report (New Reviewer)
This manuscript presents very interesting data on the concordance and discordance between systematic and targeted prostate biopsies. Though many similar studies have already been reported, what is the novelty of this study? Based on the results from Tables 3, 4 and 5, what are the possible clinical applications for future prostate biopsies? We would appreciate it if you could describe these points specifically.
Author Response
Response letter to Reviewer 3.
We thank the comments and suggestions of the Reviewer and the possibility to modify the manuscript according to them.
Comment: This manuscript presents very interesting data on the concordance and discordance between systematic and targeted prostate biopsies. Though many similar studies have already been reported, what is the novelty of this study?
Response: This study shows that systematic biopsies are needed to detect the maximum number of csPCa. The novelty of this study was the development of a predictive model of csPCa only in targeted biopsies which was helpful to avoid 14.7% of systematic biopsies missing 2.4% of csPCa, and 18.3% of prostate biopsies missing 4.6% of csPCa.
Comment: Based on the results from Tables 3, 4 and 5, what are the possible clinical applications for future prostate biopsies?
Response: Table 3 presents the multivariate analysis of candidate predictors of concordance between targeted and systematic biopsies which is the base for the development of the predictive model for csPCa only in targeted biopsies. Table 4 and 5 mainly presents the rate of csPCa detected only in systematic biopsies according to dichotomic clinical variables and those related with MRI and prostate biopsy. This data helps us to know the behavior of targeted and systematic biopsies according to those characteristics. I believe that they not have clinical applications for future prostate biopsies.

Reviewer 4 Report (New Reviewer)
Standardisation for prostate cancer diagnosis is important. The cornerstone in patients suspected for prostate cancer is mpMRI (possibly bpMRI in the near future) Quality of MRI is of utmost important in order to prevent diagnostic procedures.
What kind of MRI was performed?
Number of biopsies performed?, Biopsy schedule? How many targeted and perilesional?
What was clinically significant cancer (3+4 with or without cribriform growth pattern)
What was implication of positive biopsy found on systematic biopsy
Did patients have a unilateral lesion?
These are a number of important issues to be addressed and should be taken into account if contralateral biopsie scan be spared
See above
Author Response
Response letter to Reviewer 4.
We thank the reviewer for his comments and suggestions. Below you can find responses to the comments and modifications made in the manuscript.
Standardization for prostate cancer diagnosis is important. The cornerstone in patients suspected for prostate cancer is mpMRI (possibly bpMRI in the near future). Quality of MRI is of utmost important in order to prevent diagnostic procedures.
Response: We agree with this comment.
What kind of MRI was performed?
MpMRI was always performed. In70.5% of cases mpMRI was performed in a 3.0 Tesla magnetic strength field scan whereas in 29.5% of cases 1.5 Tesla scans was used. PI-RADS score v.2.0 was always used.
Response: We have added the point 2.2. in Material al Methods (lines 95-196) regarding the characteristics of MRI and prostate biopsies
2.2. MRI and prostate biopsy
MpMRI exams were always performed in each participant center and reported with PI-RADS v.2.0 by experienced radiologist. Pelvic phased-array surface coil was always used and 1.5 or 3.0 Tesla magnetic strength field of MRI.
MRI-TRUS fusion images was performed with cognitive or software technique. TRUS assisted prostate biopsy was performed by transrectal or transperineal route. Targeted biopsies of suspected lesions (PI-RADS >3) were performed obtaining between 2 to 6 cores, and 12-core TRUS systematic prostate biopsy was also added. Prostate biopsies were performed by experienced urologist in each center.
Biopsy material was analysed in the pathology department of each participant center by experienced pathologist using the International Society of Urologic Pathology grade group (GG) to classify PCa. CsPCa was reported when GG was 2 or higher [19]
Number of biopsies performed?, Biopsy schedule? How many targeted and perilesional?
Response: All included men underwent targeted and systematic biopsies. Targeted biopsies of suspected lesions (PI-RADS >3) obtained between 2 and 6 cores, but perilesional cores were no obtained. A 12-core TRUS systematic prostate biopsy was also performed. This is specified in the new point 2.2. added in Material and Methods (lines 99-103)
What was clinically significant cancer (3+4 with or without cribriform growth pattern)?
Response: The finding of cribriform pattern was considered a grade group 4. However, a grade group 2 without cribriform patter was considered also a csPCa. This is added in the new point 2.2 in Material and Methods (lines 104-106)
What was implication of positive biopsy found on systematic biopsy?
Response: This is interesting. The implication csPCa detection on systematic biopsy when targeted biopsies also detect csPCa usually inform a multifocality of nonvisible csPCa in MRI. However, when csPCa is only detected in systematic biopsy means two things: 1. The possibility of a false negative result of targeted biopsy due to insufficient mapping of suspicious lesion or perilesional area. 2. The possibility of only non-MRI visible csPCa. The implication is the need of performing systematic biopsies, with the objective to detect the maximum number of csPCa as currently PCa guidelines recommend. This is commented in the discussion in lines 328-343.
Did patients have a unilateral lesion? These are a number of important issues to be addressed and should be taken into account if contralateral biopsies scan be spared.
Response: No, bilateral systematic biopsy was always performed. Unfortunately, separate report of right and left systematic biopsies was not provided. It is interesting the data recently reported by Yusim et al., The prostate 23, DOI: 10.1002/pros.24585, suggesting only ipsilateral systematic biopsies. However, it must be taken into in account that this study only included men with a single unilateral suspicious lesion on MRI.

Round 2
Reviewer 2 Report (New Reviewer)
I appreciate many improvements in the new draft. But the detailed analyses should be expanded. Introduction: line 93 …” will be explored” How comes the models were not explored? Results Table 1. Add information on ISUP score. Table 2 line TRUS-MRI: vis- (vis-what?) Tables: p values are given as =0.777. delete = Table 5. Add information on ISUP grades Discussion should comment on how the Barcelonian risk score helped to reduce number of systematic biopsies (seemingly not used). How many ISUP 5 were detected only with systematic biopsies How many adverse effects grade 3-4 were seen with targeted and systematic biopsies? Main limitation is the recent recruitment of the patients. It did not allow analysis of the clinical follow up over several years. Conclusion …. without missing csPCa and missing 4.6% of csPCa. It is contradictio in adjectio.no
Author Response
Response letter to Reviewer 2 Report (Round 2)
I thank the Reviewer´s comments in this round 2.
Comment: I appreciate many improvements in the new draft. But the detailed analyses should be expanded.
Introduction:
Comment 1: line 93 …” will be explored” How comes the models were not explored?
Response: Agree. This sentence, now in lines 89-91 has been rephrased.
Results:
Comment 1: Table 1. Add information on ISUP score.
Response: This information has been added in Table 1.
Comment 2: Table 2 line TRUS-MRI: vis- (vis-what?) Tables: p values are given as =0.777. delete =
Response: I think this comment is referred to Table 3. TRUS-MRI visual. Suggested changes have been made in all tables.
Discussion:
Comment 1: Should comment on how the Barcelonian risk score helped to reduce number of systematic biopsies (seemingly not used).
Response: Effectively, the Barcelona-risk calculators can be used to avoid unnecessary mpMRI exams (BCN-RC 1), and to avoid prostate biopsies (BCN-RC 2). Avoided prostate biopsies can be targeted plus systematic in cases with low risk of csPCa, for example in cases with PI-RADS 3. However, in cases with PI-RADS < 3, only systematic biopsies would be avoided.
In this study, the BCN-RCs were not used. We refenced that participants in the present study were selected among those recruited for the validation of BCN-RCs in Catalonia.
Comment 2: How many ISUP 5 were detected only with systematic biopsies.
Response: 8 ISUP 5 were only detected in systematic biopsies (5.6%).
This is specified in lines 203-205. The ISUP-GG of csPCa detected only in systematic biopsies corresponded to GG 2 in 99 cases (69.2%), GG 3 in 21 (14.7%), GG 4 in 15 (10.5), and GG 5 in 8 (5.6%).
Comment 3: how many adverse effects grade 3-4 were seen with targeted and systematic biopsies?
Response: Unfortunately, the adverse effects of prostate biopsy were not recruited among the participants for BCN-RCs validation in Catalonia.
Comment 4: Main limitation is the recent recruitment of the patients. It did not allow analysis of the clinical follow up over several years.
Response: We agree with this comment. A small sentence emphasizing the Reviewer´s comment has been added in lines 396-397.
Conclusion:
Comment 1: …. without missing csPCa and missing 4.6% of csPCa. It is contradictio in adjectio.
Response: Agree. We rephased this sentence in lines 418-420.

Reviewer 3 Report (New Reviewer)
The conclusion to the secondary objective is not clear-cut, but overall it has been appropriately modified and is worthy of Accept.
Author Response
Response letter to Reviewer 3 Report (Round 2)
I thank the Reviewer´s comments in this round 2.
Comment 1: The conclusion to the secondary objective is not clear-cut, but overall it has been appropriately modified and is worthy of Accept.
Response: The sentence in conclusion (lines 414-417) has been modified.

Reviewer 4 Report (New Reviewer)
I do not understand what the authors mean that cribriform growth pattern is GG4?
Still I would be interested if the finding of tumour in systematic biopsy changed the clinical decision making. Theis question was not clearly addressed
Author Response
Response letter to Reviewer 4 Report (Round 2)
I thank the Reviewer´s comments in this round 2.
Comment 1: I do not understand what the authors mean that cribriform growth pattern is GG4?
Response: I´m sorry for the inconvenience. Your original comment was: “What was clinically significant cancer (3+4 with or without cribriform growth pattern)?. My proper response is that all GG 2 or higher were considered csPCa. Therefore, both cases were considered csPCa.
It is true that cribriform pattern confers a poor prognosis. Depending on the extension of cribriform pattern been primary or secondary certain pathologist classify cases as GG 4 or 5.
Comment 2: Still I would be interested if the finding of tumour in systematic biopsy changed the clinical decision making. This question was not clearly addressed.
Response: Sorry for the inconvenience. I believe the finding of “csPCa” detected only in systematic biopsy clearly change the clinical decision making since in localized tumours they will be considered with intermediate or high-risk. If an iPCa only found in systematic biopsy a consideration for active surveillance should be considered.

This manuscript is a resubmission of an earlier submission. The following is a list of the peer review reports and author responses from that submission.
Round 1
Reviewer 1 Report
The aim of this work was to evaluate the complementarity between random and targeted biopsy and to justify the necessity of both methods to detect clinically significant prostate cancer (csPCa). For this purpose, Morset and colleagues analyzed the data of 2,122 men underwent the TRUS-guided 12-core random prostate biopsy and MRI–TRUS fusion-guided targeting biopsies. Based on the number and clinico-pathologic features of the detected csPCa (n=1,021), the complementarity of both methods as well as the predictive factors were verified.
For sure, the large scale of the patient’s data used for this analysis is the particular strength of this manuscript.However, the conclusion that the random biopsies are still needed to detect "the majority of csPCa" due to the constant number of csPCa (around 6%) detectable only in random biopsy is incomprehensible or not correctly worded in my view.
The manuscript contains several limitations, such as weaknesses regarding the statistical analysis and poor writing style. The questionable calculation of complementarities and incorrect interpretation of the results lead to considerable over-interpretations of the results in the discussion and reveal the unclear benefits of this study. Moreover, a substantial part of the manuscript should be extensively rewritten/complemented.
A complementarity of two diagnostic methods should not simply be the sum of the number of csPCa that could only be detected in the respective methods as listed in the table 3,4 and 5. The clear description to estimate the complementarity of both diagnostic methods is definitely missing and the results are questionable. Regardless, the language and writing style must be improved, particularly in the results. It´s confusing and difficult to grasp the contents. Some results are descripted twice in the same paragraph (e.g. line 171-181) some others point to the wrong tables (e.g. line 183, 216 Table 3,4 and 5). The unnecessarily long sentences due to multiple subordinate clauses and commas make reading arduous.
Discussion do not provide any reasonable assumption or possible explanation of their findings / investigations. The "interpretations" and "discussions" of the findings are definitely missing. In addition, I cannot agree withthe statement in the discussion that the complementarity of random and targeted biopsies increased with the increase of PI-RADS score (line 302-305). The increased "complementarity" is observed only due to the higher sensitivity of the MRI-TRUS targeting biopsy in PCa with higher PI-RADS score (4-5) as already well known. In my view, this is more the evidence of the clear superiority of the MRI-TRUS-fusion guided biopsy by the increase of PI-RADS score and less a confirmation of the increased complementarity of both random and targeting biopsies. Their investigation mitigates the need for a random biopsy by the increase of PI-RADS more and do not strengthen the conclusion.
Taken together, the contents of the manuscript should be considered critically and reevaluated using correct statistical methods. The results and the major part of discussion should extensively be rewritten
Author Response
Reviewer 1 Comments and Suggestions for Authors:
The aim of this work was to evaluate the complementarity between random and targeted biopsy and to justify the necessity of both methods to detect clinically significant prostate cancer (csPCa). For this purpose, Morote and colleagues analyzed the data of 2,122 men underwent the TRUS-guided 12-core random prostate biopsy and MRI–TRUS fusion-guided targeting biopsies. Based on the number and clinico-pathologic features of the detected csPCa (n=1,021), the complementarity of both methods as well as the predictive factors were verified.
We thank the Reviewer for making significant comments, suggestions, and criticisms with the final objective of improving the article. This article has been rewritten and new data generated, as well as a new English proof reading performed in MDPI editing service.
- For sure, the large scale of the patient’s data used for this analysis is theparticular strength of this manuscript. However, the conclusion that the random biopsies are still needed to detect "the majority of csPCa" due to the constant number of csPCa (around 6%) detectable only in random biopsy is incomprehensible or not correctly worded in my view.
Response: Thank you for this comment. We first define the concept of complementarity between random and targeted biopsies in lines 112-113; 2.2. Outcome variable of material and methods: “it was defined as when csPCa was identified only in a random or targeted biopsy”.
To consider that 6.7% of undetected csPCa “justify or not” to stop doing random biopsies is a clinical decision in which the incidence of csPCa is also important. In fact, this 6.7% rate means that 13.7% of 1,026 detected were detected only in random biopsies.
The statistical analysis was based on correspondence analysis, based on the Chi-square Pearson test. Additionally, under the suggestion of Reviewer 2 we have developed a model to assess the discrimination ability of targeted biopsies which is now reported in the point 2.4 of results. The ROC curve is presented in Figure 1, and the number of csPCa undetected and random biopsies saved according to the threshold sensitivities with 100 to 95% in Table 6 (lines 251-272).
The expression of “the majority of csPCa” is not appropriate and we have suppressed the las sentence of simple summary (line 42). The last sentence of abstract has been also suppressed (line 63).
The manuscript contains several limitations, such as weaknesses regarding the statistical analysis a
Response: After defining the concept of complementarity in lines 112-113. We describe in lines 124-133 the statistical method to analyse the complementarity which is based on the correspondence analysis of Pearson Chi-square test with the estimated Odds ratios and 95% of CIs. We also analysed, through univariate and multivariate analysis (binary logistic regression), the association of variables to the complementarity and the predictive variables of complementarity, respectively. Now, a model to predict csPCa in targeted biopsies was developed, and finally the number of undetected csPCa and saved random biopsies is analysed, according to 100 to 95% threshold sensitivities is presented (lines 251-272)
And poor writing style
Response: I apologize for this inconvenience. A British English proofreading was performed by the Cambridge Proofreading Company. However, a new editing English language has been performed by MDPI editing service.
and incorrect interpretation of the results lead to considerable over-interpretations of the results in the discussion and reveal the unclear benefits of this study.
Response: Thank you for this comment. I agree with the reviewer that statement that “random biopsies are needed” is not appropriated. We now conclude that complementarity between random and targeted biopsies exist. This conclusion is secondary to the result of statistical analysis and the observation that some csPCa are detected only in random or targeted biopsies. A second conclusion is that “some csPCa are only detected in random biopsy”, although more times csPCa are missed in random biopsies and only detected in targeted biopsies.
Moreover, a substantial part of the manuscript should be extensively rewritten/complemented.
Response: A new editing English language has been performed by MDPI editing service, and all manuscript has been rewritten.
- A complementarity of two diagnostic methods should not simply be the sum of the number of csPCa that could only be detected in the respective methods as listed in the table 3,4 and 5. The clear description to estimate the complementarity of both diagnostic methods is definitely missing and the results are questionable.
Response: We agree with the Reviewer. Now we have developed a model, based on the logistic regression analysis, as we have explained before.
Regardless, the language and writing style must be improved, particularly in the results. It´s confusing and difficult to grasp the contents.
Response: A new editing English language has been performed by MDPI editing service.
Some results are descripted twice in the same paragraph (e.g. line 171-181) some others point to the wrong tables (e.g. line 183, 216 Table 3,4 and 5). The unnecessarily long sentences due to multiple subordinate clauses and commas make reading arduous.
Response: I apologize. All the manuscript has been reviewed and rewrote correcting many errors.
- Discussion do not provide any reasonable assumption or possible explanation of their findings/ investigations. The "interpretations" and "discussions" of the findings are definitely missing.
Response: Discussion has been rewritten and comments explaining the results have been included.
In addition, I cannot agree with the statement in the discussion that the complementarity of random and targeted biopsies increased with the increase of PI-RADS score (line 302-305). The increased "complementarity" is observed only due to the higher sensitivity of the MRI-TRUS targeting biopsy in PCa with higher PI-RADS score (4-5) as already well known. In my view, this is more the evidence of the clear superiority of the MRI-TRUS-fusion guided biopsy by the increase of PI-RADS score and less a confirmation of the increased complementarity of both random and targeting biopsies. Their investigation mitigates the need for a random biopsy by the increase of PI-RADS more and do not strengthen the conclusion.
Response: I agree with the comment. Now we have commented to right interpretation of this results reflecting that sensitivity of targeted biopsies increase with the PI-RADS category, whereas the small amount of csPCa detected only I random biopsies remains stable with the PI-RADS increase (lines 239-241) and (lines 339-348).
Taken together, the contents of the manuscript should be considered critically and reevaluated using correct statistical methods.
Response: We have improved the statistical analysis of the complementarity and now, a predictive model discriminating the ability of targeted biopsies has been included.
The results and the major part of discussion should extensively be rewritten.
Response: Discussion has been completely rewritten according to the suggestions of the Reviewers

Reviewer 2 Report
I congratulate the authors on a well-written manuscript. I have some suggestions to help improve the presentation of results and the discussion aspect.
In the results section it will be helpful if the authors could make a decision tree sort of diagram that splits the population into nodes based on variables that were found significant in the multivariable analysis and then give data about CsPCa in random and targeted biopsies (similar to table 3) for each node. This sort of presentation helps pick associations and visualize relationships.
Also, in the discussion section please elaborate on plausible scientific reasons behind the significance of each variable found important in the multivariable analysis. Say for example you already explained that the complementarity of random biopsies over targeted biopsies is the consequence of existing csPCa not visible in MRI. I can imagine that increasing age would increase the probability of more such invisible lesions. Why would complementarity be affected by other factors in the manner in which your study was observed?
Author Response
Reviewer 2 Comments and Suggestions for Authors:
I congratulate the authors on a well-written manuscript. I have some suggestions to help improve the presentation of results and the discussion aspect.
Thank you very much to this Reviewer for his comments and suggestions to improve this article.
In the results section it will be helpful if the authors could make a decision tree sort of diagram that splits the population into nodes based on variables that were found significant in the multivariable analysis and then give data about CsPCa in random and targeted biopsies (similar to table 3) for each node. This sort of presentation helps pick associations and visualize relationships.
Response: Thank you for this proposal. The development of a tree sort with the significant predictors of complementarity is very interesting to assess the possibility of saving random biopsies knowing the amount of undetected csPCa. After working and thinking about the poor results, we decided to develop a model to predict the likelihood of csPCa detection only in targeted biopsies. This model was based on the found independent predictors of complementarity. After the development of the ROC curve (Figure 1) the AUC was calculated as the specificities corresponding to the thresholds with 100, 97.5, and 95% sensitivities for csPCa. Now, the amount of csPCa undetected and random biopsies saved is known (Table 6). This development is presented in the point new 3.3. of results (lines 251-272), and commented in the new discussion.
Also, in the discussion section please elaborate on plausible scientific reasons behind the significance of each variable found important in the multivariable analysis. Say for example you already explained that the complementarity of random biopsies over targeted biopsies is the consequence of existing csPCa not visible in MRI. I can imagine that increasing age would increase the probability of more such invisible lesions. Why would complementarity be affected by other factors in the manner in which your study was observed?
Response: After the suggestion of the Reviewer 1 we have rewritten the discussion. We have also included comments about the significance of independent predictive variables and the plausible scientific reasons behind their significance (lines 274-372).
Round 2
Reviewer 2 Report
please define complimentarity in abstract
please mention which variables were finally included in the main predictive model whose ROC was analysed
please mention in one clear line in the discussion and conclusion that in which subset of patients you recommend not doing random biopsies
Author Response
We thank the comments of the Reviewer 2 and responses are below indicating changes made in the manuscript.
In inform you that manuscript has been rewrote according to the suggestion of the Editor to improve the English language
- Please define complementarity in abstract
Response: Complementarity has been changed in all manuscript by concordance and discordance concordance degree and discordance between systematic and targeted biopsies.
- Please mention which variables were finally included in the main predictive model whose ROC was analysed
Response: Included predictive variables in the model for analysing the ability of targeted to detect csPCa in targeted biopsies are now mentioned in abstract (current lines 46-49), material and methods (current lines 255-257) and discussion (current lines 358-360).
- Please mention in one clear line in the discussion and conclusion that in which subset of patients you recommend not doing random biopsies
Response: This is mentioned now in discussion (current lines 369-371) and conclusions (current lines 395-396